# Impact of an Operating Room Nurse Preoperative Dialogue on Anxiety, Satisfaction and Early Postoperative Outcomes in Patients Undergoing Major Visceral Surgery—A Single Center, Open-Label, Randomized Controlled Trial

**DOI:** 10.3390/jcm11071895

**Published:** 2022-03-29

**Authors:** Patricia Dias, Daniel Clerc, Maria Goreti da Rocha Rodrigues, Nicolas Demartines, Fabian Grass, Martin Hübner

**Affiliations:** 1School of Health Sciences Western Switzerland-HES-SO, University of Applied Sciences and Arts, 1007 Lausanne, Switzerland; patriciamarianadias@gmail.com; 2Department of Visceral Surgery, Lausanne University Hospital (CHUV), University of Lausanne (UNIL), 1011 Lausanne, Switzerland; daniel.clerc@chuv.ch (D.C.); demartines@chuv.ch (N.D.); fabian.grass@chuv.ch (F.G.); 3School of Health Sciences-HESAV, Western Switzerland-HES-SO, University of Applied Sciences and Arts, 1007 Lausanne, Switzerland; gora.darocha@hesav.ch

**Keywords:** perioperative nursing, preoperative anxiety, postoperative outcomes, perioperative nurse dialogue, surgery outcomes

## Abstract

Background: Anxiety is common before surgery and known to negatively impact recovery from surgery. The aim of this study was to evaluate the impact of a preoperative nurse dialogue on a patient’s anxiety, satisfaction and early postoperative outcomes. Method: This 1:1 randomized controlled trial compared patients undergoing major visceral surgery after a semistructured preoperative nurse dialogue (interventional group: IG) to a control group (CG) without nursing intervention prior to surgery. Anxiety was measured with the autoevaluation scale State-Trait Anxiety Inventory (STAI, Y-form) pre and postoperatively. The European Organization for Research and Treatment of Cancer (EORTC) In-Patsat32 questionnaire was used to assess patient satisfaction at discharge. Further outcomes included postoperative pain (visual analogue scale: VAS 0–10), postoperative nausea and vomiting (PONV), opiate consumption and length of stay (LOS). Results: Over a period of 6 months, 35 participants were randomized to either group with no drop-out or loss to follow-up (total *n* = 70). The median score of preoperative anxiety was 40 (IQR 33–55) in the IG vs. 61 (IQR 52–68) in the CG (*p* < 0.001). Postoperative anxiety levels were comparable 34 (IQR 25–46) vs. 32 (IQR 25–44) for IG and CG, respectively (*p* = 0.579). The IG did not present higher overall satisfaction (90 ± 15 vs. 82.9 ± 16, *p* = 0.057), and pain at Day 2 was similar (1.3 ± 1.7 vs. 2 ± 1.9, *p* = 0.077), while opiate consumption, PONV levels and LOS were comparable. Conclusion: A preoperative dialogue with a patient-centered approach helped to reduce preoperative anxiety in patients undergoing major visceral surgery.

## 1. Introduction

Experiencing the operating room environment as a patient is a very stressful moment in life [1]. About 80% of adult patients submitted to surgery report extreme levels of anxiety prior to surgery [2,3,4,5]. Patients’ preoperative concerns often stem from the uncertainty and unfamiliarity surrounding surgery [2]. Anxiety, fear and pain are known to negatively impact recovery from surgery [1]. Related psychological and physiological repercussions such as acute anxiety or reactional depression episodes vary considerably between individual patients and hence require special nursing care awareness [6]. 

The most effective interventions from the current literature are preoperative interventions using empathy and patient-centered approaches. The patient-centered approach focuses on concerns, expectations, need for information, emotional needs and life issues [3]. A combination of effective preoperative education, individualized communication and involvement of social support should be planned [1].

While a patient-centered approach might be effective to reduce patients’ preoperative anxiety in both inpatient and outpatient settings, actual evidence remains scarce in the operating room (OR) nursing field [6]. Perioperative nursing care in OR traditionally implies little contact with patients prior and after surgery [2]. For this reason, a constant development of the perioperative nurse role is promising to provide evidence-based care, grounded on validated models and scientific health research. 

The aim of this study was to explore the impact of a preoperative OR nurse dialogue on anxiety, satisfaction and early postoperative outcomes in patients undergoing major surgery. 

## 2. Methods

### 2.1. Study Design

This study was designed as a single-center, prospective, open-label, randomized controlled trial with balanced randomization (1:1). The intervention was a dedicated preoperative dialogue with a board-certified OR nurse. The control group (CG) received no intervention and followed the usual preoperative management. The randomization was performed by the investigators through a computer-generated randomization method.

This study was approved by the local IRB (CER-VD; #2019-01516) and registered in clinicaltrials.gov (NCT05237557).

### 2.2. Patients and Setting

The study population included consecutive patients undergoing elective major visceral surgery at Lausanne University Hospital (CHUV). Major surgery was defined as any esophageal, intestinal, colorectal, hepatic and pancreatic resection for benign or malignant disease and including other intra-abdominal open or laparoscopic procedures lasting more than 2 h. Further inclusion criteria were ≥18 years, hospitalization the day before surgery, and sufficient command of the French language. Exclusion criteria were emergency procedures (after unplanned admission), procedures for metabolic surgery and organ transplantation, inability to obtain informed consent or refusal and inability to follow procedures (e.g., due to language problems, psychological disorders, dementia). 

Pain management was led by the treating anesthesiologist according to predefined standards in the setting of the institutional enhanced recovery after surgery (ERAS) protocol and care maps. Details are available in previous institutional publications on pain management in this setting [7,8].

### 2.3. Intervention

The CG received routine care and procedures following current normal flow in the operating room, which consisted of a brief welcoming interview with the anesthesia nurse based on a security checklist, while the anesthetist was focusing entirely on technical and procedural aspects, followed by a waiting period until the patient was ready to enter the OR for induction of anesthesia. The intervention group (IG) received an additional dedicated and standardized 10-min interview from a specifically trained OR nurse in the welcoming facility of the OR following a patient-centered approach [3]. No extra time was spent in the welcoming space of the OR as a result of the intervention. The predefined structure is detailed in Appendix A.

Data collection started on 9 January 2020 but was suspended during the first COVID-19 breakout on 26 February 2020. Recruitment restarted on 24 April 24 and ended on 29 June 2020. 

### 2.4. Outcomes

The primary endpoint was the level of preoperative anxiety, which was assessed 5 to 10 min after the OR nurse intervention (IG) and for both comparative groups immediately before entering the OR. The anxiety level was measured with the State-Trait Anxiety Inventory Form Y (STAI-Y) [9]. This self-report instrument is a 20-item inventory, which includes measures of the state of the anxiety feeling. Anxiety was measured again in both groups on postoperative Day 2, as a secondary endpoint (Appendix A).

Patient satisfaction was measured using the survey model of the European Organization for Research and Treatment of Cancer (EORTC) In-Patsat32 [10]. The EORTC In-Patsat32 evaluated satisfaction with doctors, nurses, health service and OR experience and was translated and validated (Cronbach’s α 0.77–0.97) in several countries and populations in Europe [11]. Likert scales were used to measure individual components (0 = no satisfaction–100 = complete satisfaction) [10]. In the present study, the French edition was used.

Postoperative pain at rest was assessed until POD 4 using a visual analogue scale (VAS 0–10), while postoperative nausea and vomiting (PONV) wereassessed as a binary outcome (yes/no), defined according to the Anesthesia Research Society [12]. Ondansetron was used as the first-line antiemetic drug according to ERAS guidelines. Further outcome measures included need for postoperative opioid administration and postoperative hospital length of stay (days). This clinical routine data was readily available from a prospectively maintained institutional database. 

### 2.5. Statistical Analysis

The sample size was calculated based on the primary endpoint (preoperative anxiety level) using the sample size calculator utility [13]. Using the same STAI-Y questionnaire, adult patients demonstrated a high level of preoperative anxiety with scores around 50 (in the scale of 20 to 80) and a lower score of around 30 in patients with an intervention using the same STAI-Y scale [9]. To achieve statistical power to detect a treatment effect of 20% (difference between IG and CG scores) and with a standard deviation of 16, the sample size needed yielded 35 patients per group for a total sample size of 70 patients (power of 80%, significance level of 0.05). 

Both researchers and patients were unaware of group allocation by computer-generated randomization.

Univariate analysis was performed with Mann–Whitney *U*-test or Student *t*-test for continuous variables and Fischer’s exact test for categorical variables. A *p* value of <0.05 was considered to be statistically significant. Analyses were performed with SPSS_27 (IBM, Armonk, NY, USA) and GraphPad Prism_8.0 (GraphPad Software, Inc., La Jolla, CA, USA). 

## 3. Results

In total, 81 patients were assessed for eligibility and 70 were enrolled in the trial. Of them, 35 were randomized to the IG and 35 to the CG. There was no drop-out, and all patients entered final analysis (Figure 1). All patients were oncological patients. Demographics, perioperative pain management and surgical procedures were comparable between both groups (Table 1). Of note, 37 patients (53%) were included during the COVID-19 pandemic.

### Outcomes

Preoperative anxiety scores were statistically significantly lower in the IG (40 (IQR 33–55) vs. 61 (IQR 52–68), *p* < 0.001), while no statistically significant difference was observed for postoperative anxiety (34 (IQR 25–46) vs. 32 (IQR 25–44), *p* = 0.579, Figure 2). The COVID-19 pandemic had no impact on pre (50.2 ± 19.3 in pre-COVID-19 phase vs. 51.3 ± 13.3 during the pandemic, *p* = 0.772) and postoperative (36 ± 14 vs. 36.1 ± 13.5, *p* = 0.917) anxiety levels.

Postoperative pain and PONV levels were low, comparable throughout POD 1–4 (Figure 3), despite a trend toward less PONV in the IG. Opiates were administered postoperatively in 68.6% of patients in the IG and in 60% of patients in the CG (*p* = 0.618).

Satisfaction scores with doctors and nurses were high through all categories and similar between both groups (Figure 4). Regarding hospital services and care, the items access and comfort/cleanliness were rated higher by the IG. Finally, OR reception and comfort/cleanliness were also rated significantly higher by the IG, while overall satisfaction was similar (90 ± 15 vs. 82.9 ± 16, *p* = 0.057). Preoperative information by the operating nurse was rated similarly by patients in both group (85 ± 017 vs. 76 ± 22, *p* = 0.069). 

Median length of hospital stay was 6 (IQR 4.5–13.5) days in the intervention and 10.5 (IQR 5.5–16.5) days in the control group (*p* = 0.468).

## 4. Discussion

A standardized preoperative nursing dialogue helped decrease preoperative anxiety in patients undergoing major visceral surgery. This short and inexpensive intervention can be considered for this type of surgery and probably by extension to other types.

Nursing interventions with the aim of reducing anxiety prior to surgical interventions have been assessed in a recent systematic review and meta-analysis [14]. Eight out of nine included studies were randomized trials, and all but one used nursing interviews as anxiety-reducing intervention. In their meta-analysis including 864 patients, a −5.5 point reduction in the preoperative anxiety on the STAI scale was found, consistent with the findings of the present study. While most trials used various forms of educational and informative nursing interviews [5,15,16], others used empathic interviews aiming to address and respond to patient concerns and emotions [3] or motivational interviews aiming to set lifestyle goals (doi.org/10.17533/udea.iee.v37n2e07).

The present study used an informative interview with a patient-centered approach. Both informative and empathic interviews are important to relieve preoperative anxiety through detailed explanations, while addressing patient concerns at the same time. A single dedicated OR nurse performed all interviews, ensuring homogeneity of the intervention and specific knowledge on the procedure steps and setup. In these study conditions, limiting the intervention to one single nurse may prevent a potential bias related to personal characteristics of a specific provider. The intervention occurred in the OR welcoming space just before entering the operating room and lasted 10 min, which is less time than previously reported [3,15,16]. This approach has the benefit of being less complex and resource consuming than a dedicated planned outpatient nurse consultation. Moreover, the intervention fitted in the standard journey of surgical patients.

The present study revealed a decrease of preoperative anxiety in the intervention group. Overall postoperative satisfaction was high but similar in both groups. Nevertheless, opiate consumption, pain at Day 2, PONV and LOS remained similar in both groups. Postoperative anxiety was similar in both groups. This may be due to the fact that the intervention mainly focused on surgery-related aspects and might serve as an argument to consider repeated interviews focusing on the postoperative course. There is currently strong evidence that a dedicated preoperative nursing dialogue improves patient reported experience measures (PREMs), mainly due to decreased anxiety and improved satisfaction, as shown by previous RCTs [3,5,16]. Prerecorded educational animation videos prior to colorectal surgery may represent an alternative achieving similar results [17]. However, the effect of such interventions on perioperative outcomes such as pain and surgical recovery remains controversial. Pereira et al. reported significantly improved recovery and pain assessed at POD 1 in an ambulatory setting [3]. Their findings seem consistent with ours, but the effect may be more pronounced for ambulatory minor surgery compared to inpatient major surgery.

Enhanced Recovery After Surgery (ERAS) has become the new standard of perioperative care [18,19]. Furthermore, increased compliance to the evidence-based-multimodal system is strongly associated with better outcomes [20]. After ERAS implementation, the philosophy is to strive for additional marginal gains, provided the interventions can be easily implemented into the complex perioperative care pathway. Nevertheless, our present intervention seems to target patient-reported outcome measures (PROMs) and PREMs more specifically than clinical postoperative outcomes [21]. Correlation between PROMs/PREMs and postoperative complications, however, seems weak. The recent PATRONUS study showed no correlation between postoperative complications and the PROMs evaluated in a large cohort of patients undergoing major oncologic surgery [21]. This data underline the different health perspectives between outcomes reported by the patient (PROMs/PREMs) and postoperative morbidity, usually reported by the physicians.

Although often neglected in the past, PROMs and PREMs are increasingly recognized as relevant outcome measures in the current surgical literature beyond physician-led surgical outcomes such as morbidity, LOS or oncologic outcomes. In particular, PROMs have the advantage of identifying somehow underreported negative events [22]. PROMs as subjective endpoints are mostly assessed through surveys. Nevertheless, an increasing number of studies address PROMs to standardize and validate these outcome measures such as the recent description from the #OpenSourceResearch collaborative of a general-psycho-physical score as a core outcome after colorectal surgery [23]. Regarding PREMs, their relation to surgical outcomes and PROMs is more subtle. In their study reporting on PROMs & PREMs after hip/knee replacements and groin repairs, Black et al. reported a weak association between outcomes and experience [24]. However, higher PREMs were associated with a decreased likelihood of postoperative complications. Interestingly, the most relevant aspects linked to better outcomes were communication and trust with care teams along with higher explanation and involvement levels. This confirms the relevance of the intervention used in the present study.

Alternatives such as virtual-reality (VR) experiences may also be considered to improve preoperative education and to decrease anxiety with the potential for better resource and staff allocation. In a meta-analysis of 10 RCTs comprising 813 patients, preoperative anxiety was lower in pediatric patients after VR simulation but was unchanged in the adult population [25]. Nevertheless, these innovative approaches remain promising and deserve further consideration and investigation. 

Premedication prior to anesthesia was administered in selected cases with the aim of improving general well-being and overall patient satisfaction, reducing perioperative anxiety, reducing perioperative shivering, in line with recent evidence questioning systematic use [26]. In our institution, benzodiazepines were only given to patients with high-risk anxious episodes or a high level of anxiety or at a patient’s request. If given orally 1–2 h before surgery, they have only a small effect on cardiorespiratory function, but large doses can interfere with the speed and quality of recovery [27].

The present study has several limitations. Evident even if statistically nonsignificant, differences (type II error) for potential confounders (Table 1) between the comparative groups may have induced bias. Despite proper randomization and adequate study sample regarding the primary outcome, a larger study is needed to confirm these results. Pain and satisfaction underlie wide individual variations between different sociocultural circumstances [28], and the results of our study cannot be generalized to other settings.

In summary, a standardized preoperative nurse-patient dialogue, although not changing the postoperative outcomes, appears to be helpful in decreasing preoperative anxiety and could potentially optimize patients’ perioperative experience.

## Figures and Tables

**Figure 1 jcm-11-01895-f001:**
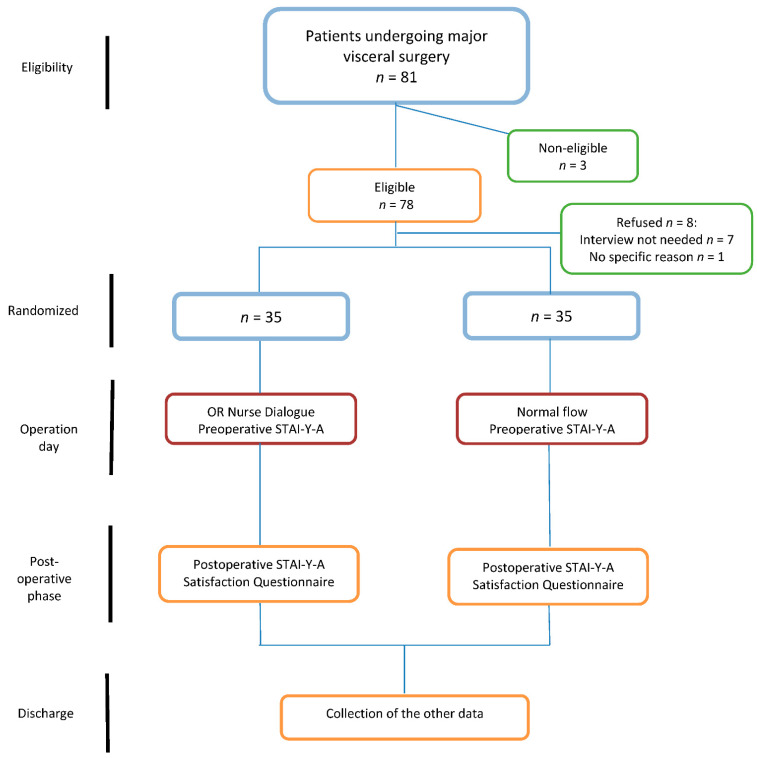
Study flowchart. STAI-Y-A: State-Trait Anxiety Inventory Form Y-A.

**Figure 2 jcm-11-01895-f002:**
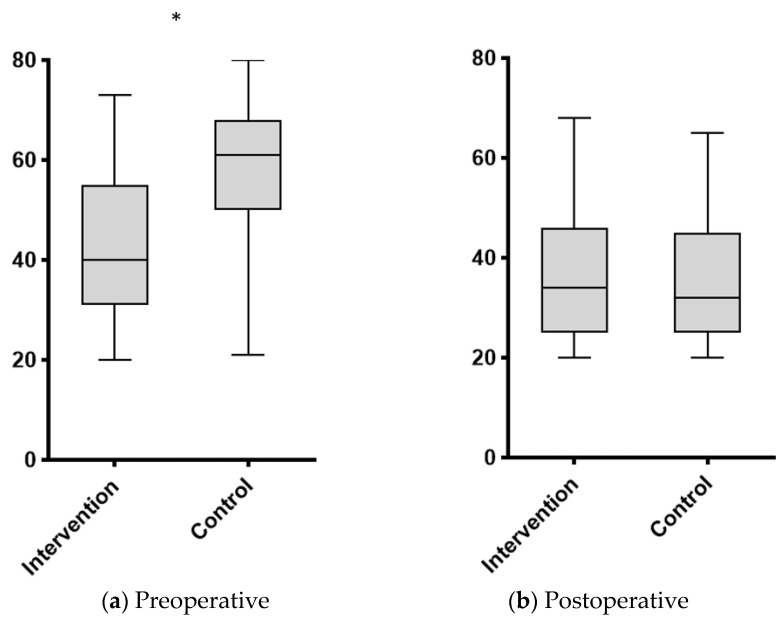
Anxiety before and after major abdominal surgery in patients with or without preoperative nurse dialogue. Whisker plots displaying: (**a**) pre-; and (**b**) postoperative anxiety levels as assessed by the State-Trait Anxiety Inventory Form Y (STAI-Y) questionnaire. ***** indicates statistical significance (*p* < 0.05). *y*-axis: STAi-Y score.

**Figure 3 jcm-11-01895-f003:**
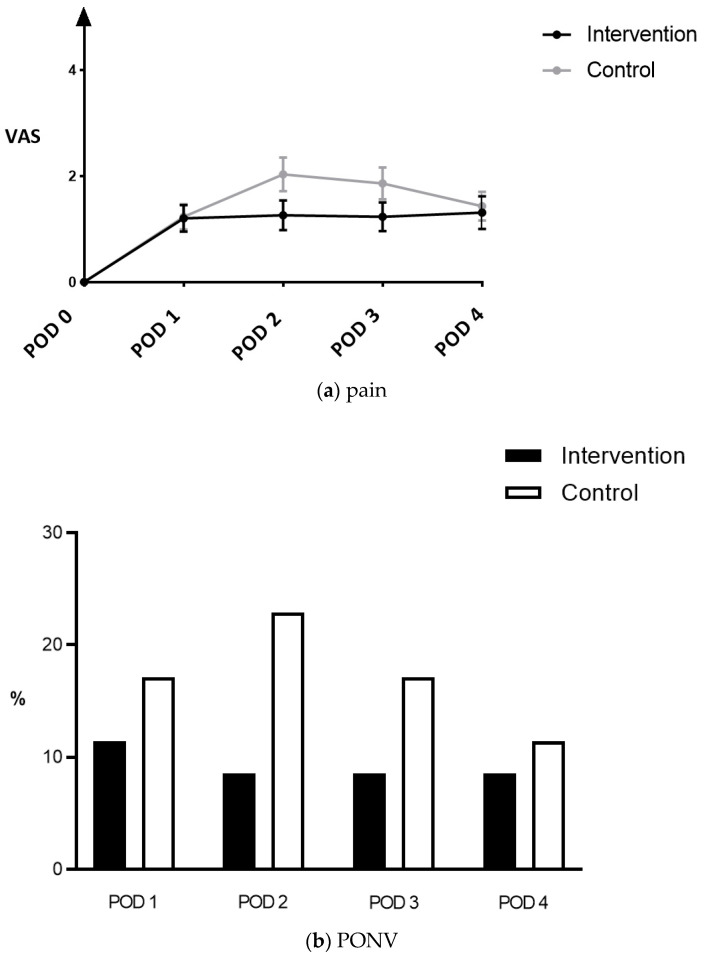
Postoperative pain and nausea: (**a**) Evolution of postoperative pain scores over time. Displayed are mean scores (dots) with S.E.M.; VAS—visual analog scale (0–10), POD—postoperative day S.E.M.—standard error of the mean; (**b**) Percentage of patients presenting with postoperative nausea in patients in the intervention group (black bars) and patients in the control group (white bars). PONV—postoperative nausea and vomiting, POD—postoperative day.

**Figure 4 jcm-11-01895-f004:**
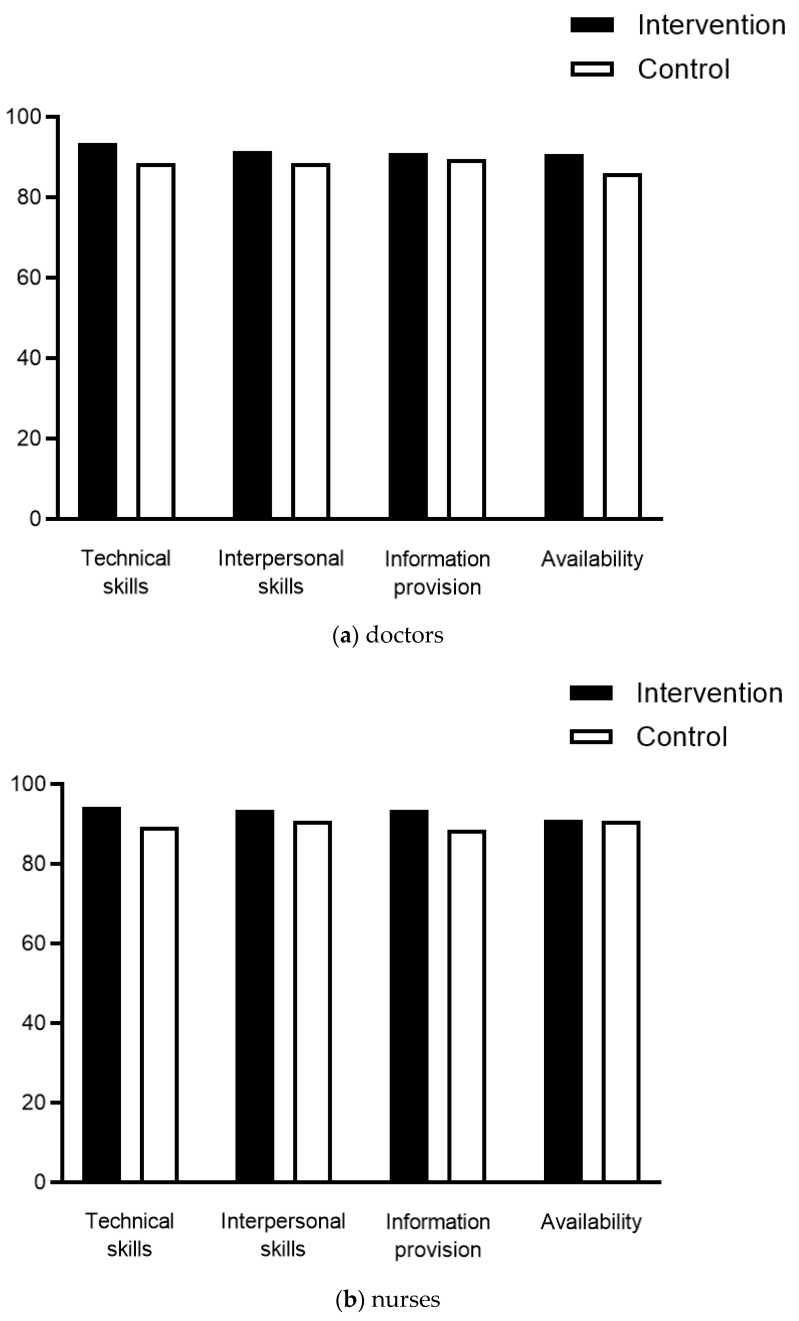
Patient satisfaction. Patient satisfaction scores (*y*-axis) (EORTC In-Patsat32 questionnaire) regarding: (**a**) doctors; (**b**) nurses; (**c**) hospital services and care; and (**d**) operating room facilities comparing patients in the intervention group (black bars) and patients in the control group (white bars). Displayed are mean scores. ***** indicates statistical significance (*p* > 0.05). HCP: Healthcare providers.

**Table 1 jcm-11-01895-t001:** Demographics and surgical details.

	Intervention Group *n* = 35	Control Group *n* = 35	Total *n* = 70	*p*-Value
Age (years, mean ± SD)	56.6 ± 14.6	60.5 ± 16.1	58.6 ± 15.4	0.285
≥70 years (%)	10 (28.6)	14 (40)	24 (34.3)	0.450
Gender (female, %)	17 (48.6)	25 (71.4)	42 (60)	0.087
ASA score ≥ 3	6 (17.1)	9 (25.7)	15 (21.4)	0.561
Premedication (%)	13 (37.1)	17 (48.6)	30 (42.9)	0.469
Peridural analgesia (%)	15 (42.9)	19 (54.3)	34 (48.6)	0.473
COVID-19 phase (%)	18 (51.4)	19 (54.3)	37 (52.9)	1.000
Type of major surgery (%)				0.699
Colorectal	18 (51.4)	15 (42.8)	33 (47.1)	
HPB	12 (34.3)	15 (42.8)	27 (38.6)	
Upper GI	3 (8.6)	5 (14.4)	8 (11.4)	
Other (Endocrine)	2 (5.7)	0	2 (2.9)	

Baseline demographic and surgical parameters of patients in the intervention and control group: ASA—American Society of Anaesthesiology; HPB—Hepatopancreaticobiliary; GI—gastrointestinal. Age is presented as mean ± standard deviation (SD). All others are frequency with percentage.

## Data Availability

The data presented in this study are available upon reasonable request from the corresponding author. The data are not publicly available due to patient privacy.

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
