# Peer review of "Impact of an Operating Room Nurse Preoperative Dialogue on Anxiety, Satisfaction and Early Postoperative Outcomes in Patients Undergoing Major Visceral Surgery—A Single Center, Open-Label, Randomized Controlled Trial"

_jcm, 2022, doi:10.3390/jcm11071895_

Round 1
Reviewer 1 Report
I think it is a very interesting study with low cost and potentially benefit the outcome of the surgical patient. The study design is clear and the conclusion is appropriate based on what the study yielded.
Please see the following comments:
- I do not see the detailed intervention from the 10-minute individual interview. What does the interview cover? It is surgery or recover explanation oriented or the anxiety-relief oriented? Please specify what is the contents of the 10-minute interview. I do not see the the Appendix 1 (LN83).
- There is minor flow in the design but I don't think it it a deal breaker as long as you explained it well. There is 10-minute individual interview in the IG, what about the CG? Do they have the same extra 10 minutes before entering the OR? How do you know the better anxiety score in IG is related to the interview not the extra 10-minute time they had to relax before the surgery?
- It is interesting that the author listed the percentage of the patients >70 yo in each group but did not make any comment. Why 70 not 65?
- To follow-up the #3, the study did not comment on postoperative delirium which probably more relevant to the preoperative anxiety.
- There is a difference(not statistically) in the incidence of PONV Figure 3. Could you comment on that?
- The manuscript will be improved if the authors provided the amount of premedication and intraoperative pain medication in both groups. As preoperative sedation, intraoperative and postoperative narcotic consumption, and the use of regional anesthetic blocks are major confounders for the postoperative outcomes.
Author Response
Reviewer #1
I think it is a very interesting study with low cost and potentially benefit the outcome of the surgical patient. The study design is clear and the conclusion is appropriate based on what the study yielded.
Thank you for your comment.
Please see the following comments:
- I do not see the detailed intervention from the 10-minute individual interview. What does the interview cover? It is surgery or recover explanation oriented or the anxiety-relief oriented? Please specify what is the contents of the 10-minute interview. I do not see the the Appendix 1 (LN83).
Thank you for comment. It has been submitted in an inappropriate section during initial submission of the manuscript. We apologize for that. It has now been added to the main document for review.
- There is minor flow in the design but I don't think it it a deal breaker as long as you explained it well. There is 10-minute individual interview in the IG, what about the CG? Do they have the same extra 10 minutes before entering the OR? How do you know the better anxiety score in IG is related to the interview not the extra 10-minute time they had to relax before the surgery?
All patients were admitted in a welcoming space of the operating room where the 10 minutes interview for patients in the IG took place. In the CG, the time frame was equivalent, but without the dedicated interview. No delay in starting the procedure was necessary due to the intervention, and IG patients had not more time to relaxcompared to the CG. These important points were specified in the methods section.
- It is interesting that the author listed the percentage of the patients >70 yo in each group but did not make any comment. Why 70 not 65?
Both cut-offs have been repeatedly used for oncological patients. Our group adheres to a cut-off of 70 years to represent the older patient population.
- To follow-up the #3, the study did not comment on postoperative delirium which probably more relevant to the preoperative anxiety.
This is an interesting point. We did not collect this data prospectively since we did not hypothesize that a nursing interview would affect the incidence of postoperative delirium.
- There is a difference (not statistically) in the incidence of PONV Figure 3. Could you comment on that?
As there was no statistical difference, we preferred to consider this as no difference, in line with the Editorial comments. However, we specified this in the results section.
- The manuscript will be improved if the authors provided the amount of premedication and intraoperative pain medication in both groups. As preoperative sedation, intraoperative and postoperative narcotic consumption, and the use of regional anesthetic blocks are major confounders for the postoperative outcomes.
Thank you for your comment. Unfortunately, we did not collect data regarding intra-operative pain medication. In line with the Editorial comments, we specified the institutional pain management strategy by referring to institutional publications. Regarding premedication, our department’s Enhanced Recovery After Surgery (ERAS) program as routine care for all patients does not consider premedication on a regular basis. Postoperative pain management is standardized through care maps, favoring multi-modal anesthesia, considering regional blocks if applicable (mainly epidural for major surgery). In this study, standard care was applied, and no pre-defined doses or opiate thresholds were set. Caring anesthesiologists were in charge of pain management in the setting of these recommendations.

Reviewer 2 Report
Impact of an operating room nurse pre-operative dialogue on anxiety, satisfaction and early postoperative outcomes in pa tients undergoing major visceral surgery – a single center, 4 open-label, randomized controlled trial 5
Patricia Dias1, Daniel Clerc2, Maria Goreti da Rocha Rodrigues3, Nicolas Demartines2, Fabian Grass2 and Martin 6 Hübner2
In general
Interesting study addressing effect on the short term of a preoperative nurse drive dialogue on anxiety, satisfaction and postoperative outcomes undergoing major surgery. This study is relevant with respect to non-pharmacological interventions to reduce perioperative anxiety and secondary outcomes. Authors used an adequate design and methodology. A point of attention is that execution of the interviews by a single nurse may induce. In general it is well written and conclusion matches the study question and chosen methodology.
Abstract:
Clearly written, no comments
Introduction
Line 41: Can authors add some examples for “Psychological and physiological repercussions underlie wide variation” to make this more clear? Now this sentence raises questions. Where do we have to think of?
Line 52: why is a constant development of the perioperative nurse role so promising? I mean, why the nurse? Is there no role for physicians?
Methods
Line 67: Patients and setting
Can having a malignant or benignant diagnosis with respect to cancer have influence on the results? If so, how was this taken into account?
Line 83: The predefined structure is detailed in appendix 1. Appendix 1 was not accessible for me as a reviewer
Line 88: The primary endpoint was the level of preoperative anxiety. To be and feel comfortable by experiencing less preoperative anxiety is of specifically patient related importance. Is this endpoint only direct preoperatively? If patients are inhospital on day -1, when did the personalized intervention take place?
Line 92: Anxiety was measured again in both groups postoperatively as a secondary endpoint. When was this done exactly?
Line 99: Postoperative pain at rest was assessed until POD 4 using a visual analogue scale. How did authors take into account other factors influencing pain scores like moving, coughing or rest; taking only one pain score may be give not adequate information.
Figure 1: 8 patients were not needed and therefore not included. How and when was determined which patients specifically were not included or maybe excluded and why were these not included? Because preoperatively, I guess, it was not known with certainty that all 70 patients would finalize their participation in the study, this raises questions. Can authors comment on that?
Tables and figures: please check all and add where missing titles and in formation to the y-axis.
Line 140-150: With respect to findings representing differences, if significantly, I suspect authors mean statistically significantly different. Please make this clear where appropriate and add this qualification where applicable. Furthermore, in the results and accessory tables, if statistically significant differences are found authors better also provide the exact numbers, which are now difficult to extract exactly form the figures.’
Line 140-142 Postoperative pain and PONV levels were low and comparable throughout POD 1-4 140 (Figure 3). Opiates were administered postoperatively in 68.6% of patients in the IG and in 60% of patients in the CG (p=0.618). How was anti-emetic drug use?
Discussion
Line 196-198: single dedicated OR nurse performed all interviews, ensuring homogeneity of the intervention and specific knowledge on the procedure steps and set-up. Can the anxiety reducing effect and other found differences in effects between IG and CG also be explained by and attributed to the characteristics of this single dedicated OR nurse? How well can the results be extrapolated to other caregivers in this light? Can authors comment on this?
Line 237-293: Regarding PREMs,its relation to surgical outcomes and PROMs is more subtle. In their study reporting on PROMs & PREMs after hip/knee replacements and groin repairs, Black et al. reported a weak association between the two outcome measures. This does not read clearly with respect to “association between the two outcome measures”. Which two outcome measures do authors mean and adapt this in the text?
Line 250-256: Premedication by means of benzodiazepines that were only prescribed for patients with high risk anxious episodes or a high level of anxiety and on patient’s request: how and how many were patients identified being at high risk? How was distribution over both patient groups and can authors comment on its effects on the results? Can authors exclude relevant effect of benzodiazepines on the primary outcomes?
Author Response
Reviewer #2
In general
Interesting study addressing effect on the short term of a preoperative nurse drive dialogue on anxiety, satisfaction and postoperative outcomes undergoing major surgery. This study is relevant with respect to non-pharmacological interventions to reduce perioperative anxiety and secondary outcomes. Authors used an adequate design and methodology. A point of attention is that execution of the interviews by a single nurse may induce. In general it is well written and conclusion matches the study question and chosen methodology.
Thank you for your comment.
Abstract:
Clearly written, no comments
Thank you for your comment.
Introduction
Line 41: Can authors add some examples for “Psychological and physiological repercussions underlie wide variation” to make this more clear? Now this sentence raises questions. Where do we have to think of?
This point was specified as requested.
Line 52: why is a constant development of the perioperative nurse role so promising? I mean, why the nurse? Is there no role for physicians?
The person-centered nursing framework model was identified as an essential foundation for health-care quality and patient safety by the WHO. It is is a model in which health-care providers are encouraged to partner with patients to co-design and deliver personalized care. Person-centered care also implies having access to one’s own nurse who is present both physically and emotionally through the entire process and who guides and follows up the patient, guaranteeing that the patient is not alone.
Methods
Line 67: Patients and setting
Can having a malignant or benignant diagnosis with respect to cancer have influence on the results? If so, how was this taken into account?
Benign diagnoses were not an exclusion criteria for the present study. Our included patients were only oncological patients (see results section), mainly because the study was performed during the COVID-19 outbreak where major surgery has been restricted to malignant surgery in our institution. This contributed to the homogeneity of the cohort but impedes comparison to benign cases.
Line 83: The predefined structure is detailed in appendix 1. Appendix 1 was not accessible for me as a reviewer
Thank you for comment. Unfortunately, it has been submitted in an inappropriate section during initial submission of the manuscript. It has now been added to the main document for review.
Line 88: The primary endpoint was the level of preoperative anxiety. To be and feel comfortable by experiencing less preoperative anxiety is of specifically patient related importance. Is this endpoint only direct preoperatively? If patients are inhospital on day -1, when did the personalized intervention take place?
We kindly refer to the methods section for a detailed description when the intervention happened.
“The CG received routine care and procedures following current normal flow on the operating room, which consisted of a brief welcoming interview with the anesthesia nurse based on a security check list, while the anesthetist was focusing entirely on technical and procedural aspects, followed by a waiting period until the patient was ready to enter the OR for induction of anesthesia. The intervention group (IG) received an additional dedicated and standardized 10-minute interview from a specifically trained OR nurse in the welcoming facility of the OR following a patient-centered approach (3). No extra time was spent in the welcoming space of the OR as a result of the intervention.”
Line 92: Anxiety was measured again in both groups postoperatively as a secondary endpoint. When was this done exactly?
Anxiety was assessed on POD 2. We specified this in the methods section.
Line 99: Postoperative pain at rest was assessed until POD 4 using a visual analogue scale. How did authors take into account other factors influencing pain scores like moving, coughing or rest; taking only one pain score may be give not adequate information.
Pain at rest is in our practice the most adequate baseline parameter, whereas pain at mobilization depends on the individual patient. Hence, pain in other situations were not prospectively collected. In our experience, pain at mobilization correlate well with pain at mobilization (Cachemaille et al., Pain Med 2020, doi 10.1093/pm/pnz156).
Figure 1: 8 patients were not needed and therefore not included. How and when was determined which patients specifically were not included or maybe excluded and why were these not included? Because preoperatively, I guess, it was not known with certainty that all 70 patients would finalize their participation in the study, this raises questions. Can authors comment on that?
8 patients refused to participate in the study. Figure 1 specifies the reasons why these patients did refuse to participate. Seven patients estimated such an interview was not beneficial to them. One patient refused to give a particular reason. Modifications were made in the figure for clarification.
Tables and figures: please check all and add where missing titles and in formation to the y-axis.
We carefully checked tables and figures.
Line 140-150: With respect to findings representing differences, if significantly, I suspect authors mean statistically significantly different. Please make this clear where appropriate and add this qualification where applicable.
We specified this where applicable. However, for the sake of brevity, we decided to display only a restricted number of pertinent p-values.
Line 140-142 Postoperative pain and PONV levels were low and comparable throughout POD 1-4 140 (Figure 3). Opiates were administered postoperatively in 68.6% of patients in the IG and in 60% of patients in the CG (p=0.618). How was anti-emetic drug use?
Anti-emetic drug use is standardized according the institutional Enhanced Recovery after Surgery (ERAS) protocol. Ondasetron was used as first line drug. This was specified in the methods section.
Discussion
Line 196-198: single dedicated OR nurse performed all interviews, ensuring homogeneity of the intervention and specific knowledge on the procedure steps and set-up. Can the anxiety reducing effect and other found differences in effects between IG and CG also be explained by and attributed to the characteristics of this single dedicated OR nurse? How well can the results be extrapolated to other caregivers in this light? Can authors comment on this?
This important point was added to the discussion section. We cannot entirely exclude a confounding effect related to the personal characteristics of the nurse performing the study, but we do think that performing the intervention in homogeneous standardized way by the same caregiver limits interpersonal bias in these study conditions.
Line 237-293: Regarding PREMs,its relation to surgical outcomes and PROMs is more subtle. In their study reporting on PROMs & PREMs after hip/knee replacements and groin repairs, Black et al. reported a weak association between the two outcome measures. This does not read clearly with respect to “association between the two outcome measures”. Which two outcome measures do authors mean and adapt this in the text?
The text has been adapted for clarification.
Line 250-256: Premedication by means of benzodiazepines that were only prescribed for patients with high risk anxious episodes or a high level of anxiety and on patient’s request: how and how many were patients identified being at high risk? How was distribution over both patient groups and can authors comment on its effects on the results? Can authors exclude relevant effect of benzodiazepines on the primary outcomes?
Unfortunately, we don’t dispose of these data. Premedication was not systematically used on a regular basis in line with ERAS guidlines. As the present study has a randomized controlled design, normal distribution of this particular confounder among the groups can be reasonably expected.

Round 2
Reviewer 1 Report
Thanks for addressing my comments and questions.
Here are some minor comments, please see the attachment.

Author Response
Thank you for your comments and through review of our manuscript.
Comments made in the revised version (R1) have been accepted and the manuscript has been modified accordingly.
